# Phylogeny and Toxin Profile of Freshwater Pufferfish (Genus *Pao*) Collected from 2 Different Regions in Cambodia

**DOI:** 10.3390/toxins12110689

**Published:** 2020-10-30

**Authors:** Hongchen Zhu, Akinori Yamada, Yui Goto, Linan Horn, Laymithuna Ngy, Minoru Wada, Hiroyuki Doi, Jong Soo Lee, Tomohiro Takatani, Osamu Arakawa

**Affiliations:** 1Graduate School of Fisheries and Environmental Sciences, Nagasaki University, 1-14, Bunkyo-machi, Nagasaki 852-8521, Japan; zhc957286316@hotmail.com (H.Z.); ayamada@nagasaki-u.ac.jp (A.Y.); miwada@nagasaki-u.ac.jp (M.W.); taka@nagasaki-u.ac.jp (T.T.); 2Faculty of Fisheries, Nagasaki University. 1-14, Bunkyo-machi, Nagasaki 852-8521, Japan; gtoui555@gmail.com; 3University of Kratie, Orussey District, Kratie Province, Cambodia; hornlinan189@gmail.com (L.H.); ngy_mithuna@yahoo.com (L.N.); 4Nifrel, Osaka Aquarium Kaiyukan. 2-1, Senribanpakukoen, Suita, Osaka 565-0826, Japan; doi@kaiyukan.com; 5College of Marine Science, Gyeongsang National University, 2, Tongyeonghaean-ro, Tongyeong, Kyungnam 53064, Korea; leejs@gnu.ac.kr

**Keywords:** Cambodia, pufferfish, *Pao*, phylogenetic analysis, tributyltin-binding protein type 2 (TBT-bp2), saxitoxin (STX)

## Abstract

The species classification of Cambodian freshwater pufferfish is incomplete and confusing, and scientific information on their toxicity and toxin profile is limited. In the present study, to accumulate information on the phylogeny and toxin profile of freshwater pufferfish, and to contribute to food safety in Cambodia, we conducted simultaneous genetic-based phylogenetic and toxin analyses using freshwater pufferfish individuals collected from Phnom Penh and Kratie (designated PNH and KTI, respectively). Phylogenetic analysis of partial sequences of three mitochondrial genes (cytochrome *b*, 16S rRNA, and cytochrome *c* oxidase subunit I) determined for each fish revealed that PNH and KTI are different species in the genus *Pao* (designated *Pao* sp. A and *Pao* sp. B, respectively). A partial sequence of the nuclear tributyltin-binding protein type 2 (TBT-bp2) gene differentiated the species at the amino acid level. Instrumental analysis of the toxin profile revealed that both *Pao* sp. A and *Pao* sp. B possess saxitoxins (STXs), comprising STX as the main component. In *Pao* sp. A, the toxin concentration in each tissue was extremely high, far exceeding the regulatory limit for STXs set by the Codex Committee, whereas in *Pao* sp. B, only the skin contained high toxin concentrations. The difference in the STX accumulation ability between the two species with different TBT-bp2 sequences suggests that TBT-bp2 is involved in STX accumulation in freshwater pufferfish.

## 1. Introduction

Pufferfish of the family Tetraodontidae contain tetrodotoxin (TTX) and/or saxitoxins (STXs), but the toxin ratio differs depending on the genus or species. Marine pufferfish of the genus *Takifugu* inhabiting the coastal waters of Japan and *Dichtomyctere* (formerly known as *Tetraodon*) living in the brackish waters of Thailand have TTX as their main toxin component [1,2,3], whereas freshwater pufferfish of the genera *Pao* and *Leiodon* (both formerly known as *Tetraodon*), which inhabit Southeast Asian countries, generally have only STXs [4,5,6,7]. On the other hand, *Sphoeroides* pufferfish from Florida, and *Arothron* and *Canthigaster* pufferfish from the Philippines, Japanese coastal waters, or the Caribbean Sea have both TTX and STXs [8,9,10,11,12]. The toxin ratio varies depending on the species, but STXs are dominant in many cases. Marine pufferfish of the genus *Takifugu* may have small amounts of STXs in addition to TTX [13,14,15].

Many studies examining the toxification mechanism of pufferfish with TTX revealed that pufferfish do not biosynthesize TTX, but accumulate it through the food chain that starts with TTX-producing bacteria [1]. The accumulated TTX is gradually eliminated if supply via toxic food is not continued [16]. We recently conducted various in vivo TTX administration experiments using nontoxic pufferfish artificially raised with nontoxic diets, and found that the internal TTX kinetics in pufferfish are unique: intestine → liver → skin/ovary, and change as the pufferfish grows and matures [17,18,19,20,21,22]. The pufferfish STX- and TTX-binding protein (PSTBP) isolated from the plasma of *Takifugu pardalis* by Yotsu-Yamashita et al. [23] is thought to be involved in the internal TTX kinetics, i.e., the absorption, transportation, and accumulation of TTX inside the pufferfish body [24]. PSTBP homologous proteins are widely distributed in TTX-bearing toxic pufferfish of the genera *Takifugu* and *Arothron*, but have not been detected in common fish or in nontoxic species of pufferfish [25,26,27].

The accumulation mechanism of STXs in pufferfish, however, remains unclear, but is presumed to be exogenous via the food chain that starts with STX-producing dinoflagellates in marine environments and STX-producing cyanobacteria in freshwater environments [6,9,11,28]. In a recent in vivo TTX/STX administration experiment using nontoxic cultured pufferfish, Gao et al. [29] found that *T. pardalis*, which naturally contains TTX, selectively accumulates TTX, and *Pao suvattii*, which naturally contains STXs, selectively accumulates STXs. In other words, the ratio of TTX/STX in pufferfish appears to depend more strongly on the inherent TTX/STX selectivity of the pufferfish than the prevalence of TTX/STX in prey organisms, but the molecular mechanisms underlying the toxin selectivity remain to be elucidated. Our ongoing genome and transcriptome analyses indicate that the three freshwater pufferfish species, *Pao abei*, *P. suvattii*, and *Leiodon cutcutia*, do not have the PSTBP gene (unpublished data), suggesting that PSTBP is not involved in STX accumulation or selectivity of the genus *Pao*.

Several species of freshwater pufferfish inhabit the rivers and lakes of Cambodia. These pufferfish are considered delicious and there have been many cases of poisoning due to the ingestion of pufferfish caught with other generally edible freshwater fish. According to the Ministry of Health in Cambodia, at least seven poisoning incidents occurred due to freshwater pufferfish consumption from 2017 to 2019, with more than 40 people poisoned and 5 deaths. The Codex Committee on Fish and Fishery Products has set the regulatory limit for STXs as 0.8 mg STX·diHCl eq/kg edible tissue [30], but there is no strict food safety standard for the edible use of freshwater pufferfish in Cambodia. This may be partly due to the fact that the species classification of Cambodian indigenous pufferfish is incomplete and quite confusing, and the limited scientific information on the toxicity and toxin profile. In the present study, to accumulate genetic information on the taxonomy of freshwater pufferfish and the TTX/STX distribution profiles in pufferfish of the family Tetraodontidae, and to contribute to food safety in Cambodia, we conducted simultaneous genetic-based phylogenetic and toxin analyses using freshwater pufferfish specimens collected from two different regions in Cambodia. Moreover, to explore candidate molecules other than PSTBP involved in the absorption, transportation, and accumulation of STXs in freshwater pufferfish, a partial sequence of a putative origin of the molecular evolution of PSTBP [31,32], tributyltin-binding protein type 2 (TBT-bp2), and four commonly used genes (cytochrome *b* [CYTB], 16S rRNA, cytochrome *c* oxidase subunit I [COI], and rhodopsin) were used as targets in the phylogenetic analysis.

## 2. Results

### 2.1. Phylogeny

A total of 16 freshwater pufferfish were sampled from small lakes near Phnom Penh (PNH) and Kratie (KTI) (Figure 1 and Figure 2, and Table A1). Partial sequences of the three mitochondrial genes, CYTB, 16S rRNA, and COI, were determined for each sample and phylogenetically analyzed for taxonomic identification. Phylogenetic analysis of 870-bp CYTB sequences showed that the PNH and KTI sequences each formed a tight cluster together with *Pao cochinchinensis* (AB741980.1) or *Pao baileyi* (AB741978.1) (Figure 3). Similar results were obtained from analyses of concatenated datasets of CYTB+16S rRNA (555 bp) and CYTB+COI (621 bp) (Figure A1 and Figure A2). Considering the genetic variation of *P. suvattii* in Figure 3, each cluster seems to represent a single species, though the intraspecific variation in the genus *Pao* might be unexpectedly large (see *Pao leiurus* and *P. cochinchinensis* in Figure 3, Figure A1, and Figure A2). Hereafter, we thus refer to the samples from PNH and KTI as *Pao* sp. A and *Pao* sp. B, respectively.

A 492 to 778 bp region of the nuclear rhodopsin gene was directly sequenced from the polymerase chain reaction (PCR) product of each sample. As no double peaks were observed in the Sanger sequence electropherograms, the sequenced regions were considered to be homogeneous in all the samples. Furthermore, there was no sequence variation between any pair of samples. Compared with sequences of other *Pao* species in the database, all the mutations found were synonymous, and the amino acid sequence was identical among the *Pao* species examined.

A 298 to 536 bp region of the nuclear TBT-bp2 gene was determined for all the samples except KTI-3 by direct sequencing of the PCR products. We observed double peaks at several nucleotide positions in the Sanger sequence electropherograms of almost all the samples, indicating that the TBT-bp2 gene had multiple alleles (Table A2). Allele types were apparently similar between samples of *Pao* sp. A and *Pao* sp. B. Compared with the allele types found in *Pao* sp. A and *Pao* sp. B, there was no sample in which the allele types could be represented as a hybridization of *Pao* sp. A and *Pao* sp. B. An ML phylogenetic analysis clearly separated the samples into PNH and KTI, namely *Pao* sp. A and *Pao* sp. B, as in the case of the mitochondrial genes (Figure 4). Amino acid sequences also differed between *Pao* sp. A and *Pao* sp. B. Note that although we tried to obtain longer sequences, PCR amplifications were unsuccessful, probably due to the presence of microsatellite-like sequences in neighboring introns.

### 2.2. Toxin Profile

Toxins were extracted from the skin, muscle, liver, and gonads of each freshwater pufferfish. Analysis of the extracts by high-performance liquid chromatography with post-column fluorescence derivatization (HPLC-FLD) for STXs [12,33] revealed that the tissues of each individual, except for one or two tissues of KTI-1 and KTI-2, contained STXs comprising STX as the main component and decarbamoylSTX (dcSTX) as a minor component (typical chromatograms are shown in Figure A3). NeoSTX was also contained in some tissues as a minor component, but no other known STX components, such as C toxins and gonyautoxins (GTXs), were detected. When the same extracts were analyzed for TTX by liquid chromatography tandem mass spectrometry (LC-MS/MS) [12,34], no TTX was detected at all.

The toxin concentration (converted value to mg STX·diHCl eq/kg) in each tissue from the pufferfish is shown in Figure 5 and Table A1. The toxin concentrations of *Pao* sp. A were generally high and far above the regulatory limit for STX (0.8 mg STX·diHCl eq/kg) in all tissues. The mean toxin concentration in the ovary (58.7 mg STX·diHCl eq/kg) was the highest, followed by the skin (40.8 mg STX·diHCl eq/kg). The mean toxin concentration in the liver (23.1 mg STX·diHCl eq/kg) was lower than that in the skin, although in some individuals, the toxin concentration in the liver exceeded that in the skin. The mean toxin concentration in the muscle (22.3 mg STX·diHCl eq/kg) was the lowest, but still almost 28-fold higher than the regulatory limit. In contrast, the toxin concentrations of *Pao* sp. B were generally much lower than those of *Pao* sp. A. The mean toxin concentration in the skin (6.0 mg STX·diHCl eq/kg) was the highest, followed by the ovary (1.3 mg STX·diHCl eq/kg). The mean toxin concentrations in the testis, liver, and muscle were below the regulatory limit (0.5–0.6 mg STX·diHCl eq/kg), but some individuals had concentrations that slightly exceeded the regulatory limit.

The total amount of toxin per individual is shown in Figure 6. As described above, the toxin concentration in each tissue was much higher in *Pao* sp. A than in *Pao* sp. B, but due to the larger individual size of *Pao* sp. B over *Pao* sp. A (Table A1), the total toxin amount of KTI-3–5 (172–417 µg STX·diHCL eq/individual) was generally comparable to that in *Pao* sp. A individuals (122–463 µg STX·diHCL eq/individual) other than PNH-1, in which the toxin concentration in the skin was exceptionally low. The toxin distribution profile in the body, however, differed between *Pao* sp. A and *Pao* sp. B. In *Pao* sp. A, in addition to the skin, the amount of toxin in the muscle and ovary was high, whereas in *Pao* sp. B, most of the toxin amount in the body was accounted for by the skin, irrespective of the sex.

The toxin composition in each tissue per species is shown in Figure 7. In *Pao* sp. B, the toxin concentrations in the muscle, liver, and gonads were very low, and neoSTX concentrations were below the limit of quantification. Other than this, however, there was no considerable difference between *Pao* sp. A and *Pao* sp. B. In each tissue, most of the toxins were accounted for by STX. In the whole body, both *Pao* sp. A and *Pao* sp. B contained 6% to 7% neoSTX and 3% to 4% dcSTX as the minor components, in addition to the main component STX (~90%).

## 3. Discussion

The present study revealed that freshwater pufferfish collected from two regions of Cambodia, Phnom Penh and Kratie, are different species belonging to the genus *Pao* (*Pao* sp. A and *Pao* sp. B, respectively), and both possess STXs, with *Pao* sp. A having high toxicity and *Pao* sp. B having low toxicity.

The CYTB analysis suggests that *Pao* sp. A is the same species as *P. cochinchinensis* (AB741980.1), whereas CYTB sequences of other *P. cochinchinensis* specimens (AP011925.1, JQ681922.1) are distantly related to *Pao* sp. A (Figure 3), which makes it difficult to identify species with DNA barcoding methods. According to Igarashi et al. [35], at least one of the CYTB sequences is likely an error resulting from the misidentification of specimens. Likewise, *Pao* sp. B is closely related to *P. baileyi* as well as to *P. abei* and *P. leiurus* (KF667490.1, GU057266.1) (Figure 3), and could be the same species as one of these. The phylogenetic positions of *P. leiurus*, however, largely differ among specimens (Figure 3). As discussed in Igarashi et al. [35], these situations are most likely due to similarities in the morphologic characteristics, geographic distribution, and ecology of the *Pao* species. Our mitochondrial gene analyses indicated the presence of actual and/or potential genetic boundaries of species (i.e., species-level clusters in the phylogenetic trees), suggesting that an accumulation of genetic information could contribute to resolve the taxonomic status and relationships among *Pao* species. On the other hand, if crossbreeding occurred in the *Pao* species, an introgression of the mitochondrial genome might have caused such discordance between morphologic species identification and mitochondrial DNA phylogeny. Crossbreeding is frequently observed in the marine *Takifugu* pufferfish species (e.g., Takahashi et al., 2017 [36]), and an introgression of the mitochondrial genome from the freshwater *Rhinogobius* goby species was reported [37]. As demonstrated by the analysis of the TBT-bp2 gene, this possibility should be carefully considered in parallel to taxonomic reexamination, though no evidence was found for the presence of crossbreeding between *Pao* sp. A and *Pao* sp. B.

The toxin concentration in each tissue differed greatly between *Pao* sp. A and *Pao* sp. B: it was generally much higher in *Pao* sp. A than in *Pao* sp. B. Ngy et al. [7] examined the toxicity of two Cambodian freshwater pufferfish species, *Pao turgidus* and *Pao* sp., during the rainy and dry seasons, and reported that only the skin and ovary of *P. turgidus* were toxic in both seasons. The highest toxicity scores of the skin and ovary were 37 mouse unit (MU)/g (5.5 mg STX·diHCl eq/kg) and 27 MU/g (4.0 mg STX·diHCl eq/kg), respectively, which indicate low levels of toxicity, similar to *Pao* sp. B. Ngy et al. [7] did not conduct a genetic analysis, and while there is some uncertainty in species identification, it is assumed that several species of low-toxicity freshwater pufferfish inhabit Cambodia. The Bangladeshi *L. cutcutia* (highest toxicity score of the most toxic tissue, the skin, 20 MU/g (3.0 mg STX·diHCl eq/kg)) [6] could also be considered a low-toxicity species, similar to *Pao* sp. B. According to Kungsuwan et al. [4], the Thai freshwater pufferfish *P. leiurus* and *P. suvattii* are both highly toxic, with the highest mean toxicity scores of the skin, muscle, liver, and ovary ranging from 61 to 109 MU/g (9.1–16 mg STX·diHCL eq/kg) in *P. leiurus* and 42 to 200 MU/g (6.3–30 mg STX·diHCL eq/kg) in *P. suvattii.* The toxin concentration of *Pao* sp. A is comparable to or even higher than that of *P. leiurus*/*P. suvattii*, and this species is one of the most highly toxic species among freshwater pufferfish.

The total toxin amount per fish was generally similar between *Pao* sp. A and *Pao* sp. B, with the highest score exceeding 400 µg STX·diHCL eq/individual in both species. As the minimum lethal dose for humans is approximately 400–1000 µg STX·diHCL eq [38,39], ingestion of one–two whole bodies of these freshwater pufferfish can cause death by poisoning. Therefore, both species should be considered extremely dangerous to eat. In *Pao* sp. B, however, the majority of the harbored toxin was distributed in the skin, and the toxin concentrations of the other tissues were below or only slightly above the regulatory limit. Therefore, at least with the individuals examined in this study, removing the skin may decrease the possibility of poisoning. In addition to individual differences, regional and seasonal variations are observed in pufferfish toxicity [1,40]. Careful accumulation of data on the phylogeny and toxin profile of freshwater pufferfish, with such considerations in mind, may help to establish a taxonomic system linked with toxicity profiles and the presentation of species that can be consumed by removing toxic parts and other treatments.

In both *Pao* sp. A and *Pao* sp. B, the main toxin was STXs comprising STX as the main component, and neoSTX and dcSTX as the minor components, and TTX was not detected at all. The toxin of *P. turgidus* from Cambodia and *Pao* freshwater pufferfish from Thailand is similarly composed of STX-dominated STXs and does not contain TTX [4,5,7]. There are some exceptional cases, such as the case of the Bangladeshi *L. cutcutia* in which a unique STX derivative (carbamoyl-*N*-methylSTX) is the main component [6,41], but even in the case of marine pufferfish that simultaneously possess STXs and TTX, the main component of STXs is often STX [9,11,13,14,15]. The origins of STXs in Floridian *Sphoeroides* and Philippine *Arothron* marine pufferfish are considered to be STX-producing dinoflagellates *Pyrodinium bahamense* and *P. bahamense* var. *compressum*, respectively, based on similarities in the toxin composition [8,9]. Although the origin of STXs in freshwater pufferfish is presumed to be STX-producing cyanobacteria [6], little is known about the toxicity of cyanobacteria in the freshwater environments of Cambodia, and the origin of STXs in Cambodian freshwater pufferfish remains to be elucidated.

The STX distribution profiles clearly differed between *Pao* sp. A and *Pao* sp. B. In contrast to TTX, of which the absorption, transportation, and accumulation in marine pufferfish have been studied with respect to candidate molecules involved in these processes (i.e., PSTBP), such molecules for STXs remain unknown regardless of the marine or freshwater pufferfish species. PSTBP genes in marine pufferfish obviously evolved as a result of the fusion of two TBT-bp2 genes at the sixth and first exons, and the sequences remain quite similar to TBT-bp2 genes on the same genome (Hashiguchi et al. 2015 [32]; Yamada et al., unpublished). TBT-bp2 was originally identified as a protein that binds the environmental toxin TBT in the fish body [31]. These findings together suggest that the TBT-bp2 gene could be one of the molecules involved in the kinetics of STX in the pufferfish body. As expected from the observed difference in the STX distribution profiles between *Pao* sp. A and *Pao* sp. B, we found different amino acid sequences of TBT-bp2 between *Pao* sp. A and *Pao* sp. B, which could result in interspecific differences in the structure and function of TBT-bp2. Given the identical amino acid sequence of the rhodopsin gene in *Pao* species, including *Pao* sp. A and *Pao* sp. B, the TBT-bp2 gene might be one of the genes that determines the physiologic and ecologic characteristics of the host. A more comprehensive analysis of the expression, distribution, and function of PSTBP/TBT-bp2 isoforms coupled with TTX/STX distribution profiles could provide a better understanding of the molecular mechanisms and evolutionary processes of TTX/STX accumulation in Tetraodontidae.

## 4. Materials and Methods

### 4.1. Pufferfish Specimens

In November 2019, 11 (PNH-1–11) and 5 (KTI-1–5) freshwater pufferfish were sampled from a small lake connected by a channel to the Mekong River near Phnom Penh and Kratie, Cambodia, respectively (Figure 1 and Figure 2, and Table A1). The specimens were transported alive to the laboratory at the University of Kratie, where PNH-1–3 were dissected to obtain the skin (including the fins), muscle, liver, and gonads. Each tissue was placed in a plastic bag, frozen, and transported to the laboratory at Nagasaki University. The other specimens were frozen as whole bodies and transported to the laboratory at Nagasaki University, where they were similarly dissected. Each tissue was stored at −80 °C until DNA extraction or toxin quantification.

### 4.2. DNA Extraction, PCR Amplification, and Sequencing

Total DNA was extracted from an approximately 5 × 5 mm piece of fin using a standard DNA lysis solution containing proteinase K. The 3 mitochondrial genes, CYTB, 16S rRNA, and COI, and 2 nuclear genes, rhodopsin and TBT-bp2, were PCR-amplified using the following primers: cytball-Fl and cytball-Rl for CYTB [35], 16sar-L and 16sar-H for 16S rRNA [42], FF2d and FR1d for COI [43], Rh193, Rh545, and Rh1039r for rhodopsin [44], and TBTBP2EX1F: 5′-AAC CAG CGC TKC TSC TGC TG-3′ and TBTBP2EX4R: 5′-TTC TCC TCT GTC AGG ACT CC-3′ for TBT-bp2 (this study). The PCR amplification was carried out in a reaction volume of 25 μL containing 12.5 μL of 2×Ampirect Plus, 0.125 μL of BIOTAQ DNA Polymerase (TaKaRa Bio Inc., Kusatsu, Japan), 500 nM each of forward primer and reverse primer, 10 μL of 10-fold diluted DNA, and the remaining volume made up by nuclease-free water. The PCR conditions were as follows: initial denaturation at 94 °C for 2 min, 35 cycles of amplification with each cycle containing 94 °C for 30 s, 52 °C for 40 s (16S rRNA, COI) or 50 °C for 40 s (CYTB, rhodopsin) or 60 °C for 40 s (TBT-bp2), 72 °C for 1 min (16S rRNA, COI), 2 min (CYTB, rhodopsin, TBT-bp2), and a final extension at 72 °C for 10 min. The amplicons were purified using ExoSAP-IT (Thermo Fisher Scientific Inc., Waltham, USA) and directly sequenced by outsourcing (FASMAC Co., Ltd., Atsugi, Japan). GenBank/EMBL accession numbers are from LC581801 to LC581879.

### 4.3. Phylogenetic Reconstructions

The sequences determined were checked and trimmed manually. In the case of the 2 nuclear genes, the Sanger sequence electropherograms were carefully examined for clear double peaks with comparable height, which reflects allelic heterogeneity at the locus. These sites were coded with IUPAC ambiguity codes in the sequence alignments. For the mitochondrial genes, a couple of previous studies [35,45] published CYTB sequences of closely related species with 16S rRNA or COI sequences. Phylogenetic reconstructions using Bayesian inference (BI) and maximum likelihood (ML) methods were conducted for CYTB and the 2 concatenated datasets, CYTB+16S rRNA, and CYTB+COI. Best fit models of nucleotide substitution for BI analyses by MrBayes 3.2.7 [46] were inferred using Kakusan 4 [47], and the GTR+I+G model was used for ML analyses by FastTree 2 [48]. In the BI analyses, 4 runs of 5 million generations were performed with 4 chains each, trees were sampled at 1000-generation intervals, and the first 25% of the trees were discarded as burn-in. In the ML analyses, bootstrap values were calculated using 1000 trees generated by SEQBOOT in the PHYLIP Package 3.698 [49], and the consensus tree was obtained by CompareToBootstrap.pl in FastTree 2 [50]. For TBT-bp 2, as the MrBayes analysis did not provide ambiguity codes, only ML analysis was conducted as described above, but with the JC+CAT model.

### 4.4. Toxin Quantification

Each tissue of the pufferfish was extracted with 0.1 M HCl, passed through an HLC-DISK membrane filter (0.45 µm, Kanto Chemical Co., Inc., Tokyo, Japan), and subjected to HPLC-FLD for STXs and LC-MS/MS for TTX according to the previously reported methods [12]. The reference materials of C1, C2, GTX1-4, and decarbamoylGTX2,3 were provided by the Japan Fisheries Research and Education Agency, and neoSTX, dcSTX, and STX purified from the toxic crab *Zosimus aeneus* [51] and crystalline TTX (Nacalai Tesque, Inc., Kyoto, Japan) were used as external standards. The limit of detection and limit of quantification for STXs were 0.001–0.007 nmol/mL (*S*/*N* = 3) and 0.003–0.02 nmol/mL (*S*/*N* = 10), and for TTX, 0.0009 nmol/mL (*S*/*N* = 3) and 0.003 nmol/mL (*S*/*N* = 10), respectively.

Based on the concentrations of STX, neoSTX, and dcSTX in each tissue (*s*, *n*, and *d* nmol/g, respectively), the toxin concentration in each tissue (*c* mg STX·diHCl eq/kg) was calculated with the toxicity equivalence factors (STX = 1, neoSTX = 2, and dcSTX = 0.5) [30] and molecular weight of STX·diHCl (372.211) using the following equation; *c* = 372.211 × (1*s* + 2*n* + 0.5*d*) × 10^−3^.

## Figures and Tables

**Figure 1 toxins-12-00689-f001:**
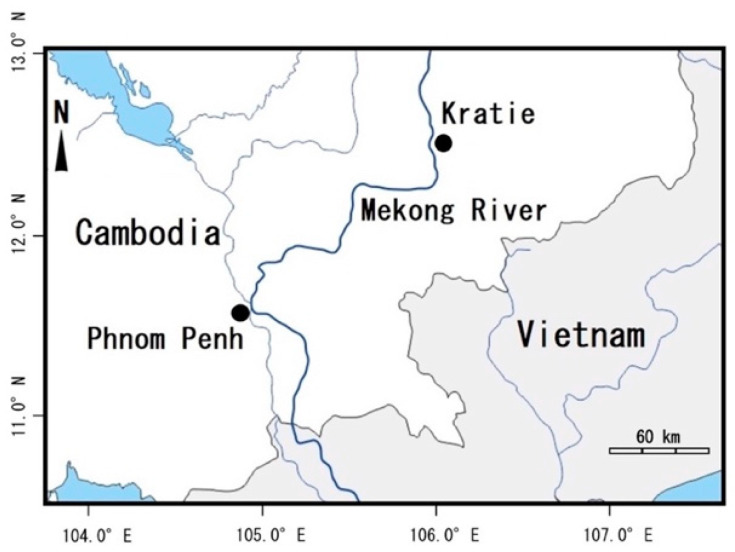
Map showing the pufferfish sampling locations in Cambodia.

**Figure 2 toxins-12-00689-f002:**
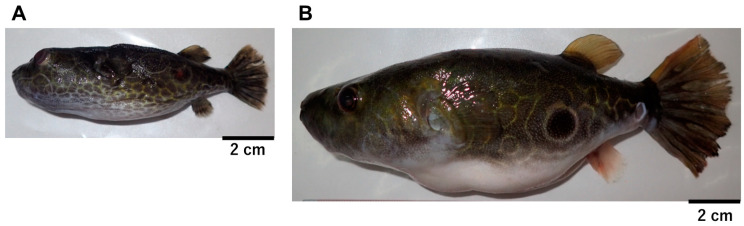
Typical pufferfish collected from Phnom Penh (**A**, PNH-1) and Kratie (**B**, KTI-2).

**Figure 3 toxins-12-00689-f003:**
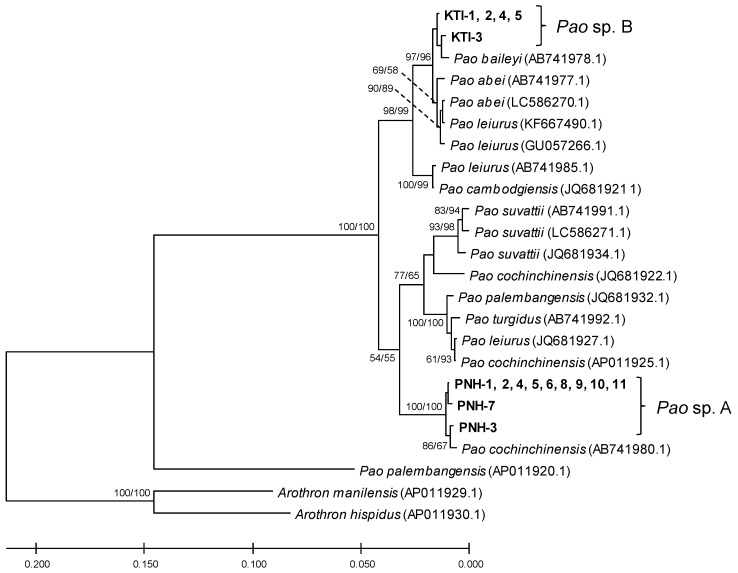
Phylogenetic tree of *Pao* species constructed by Bayesian inference (BI) and maximum likelihood (ML) based on the CYTB gene. The BI tree is shown. Node numbers are Bayesian posterior probabilities (left) and ML bootstrap values (right).

**Figure 4 toxins-12-00689-f004:**
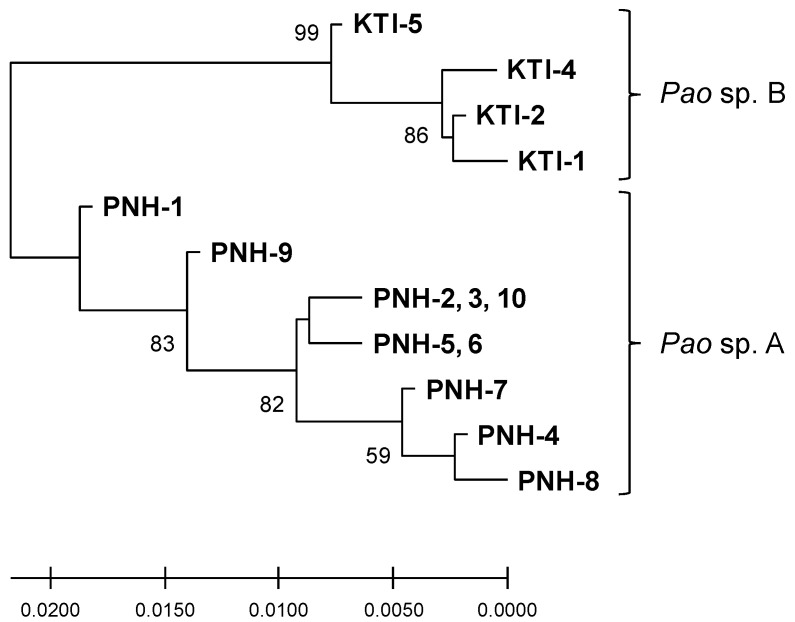
Inter-sample relationships of the TBT-bp2 gene using maximum likelihood (ML). A midpoint rooted tree is shown. Node numbers are ML bootstrap values. KTI-3 and PNH-11 were not included due to their shorter sequences.

**Figure 5 toxins-12-00689-f005:**
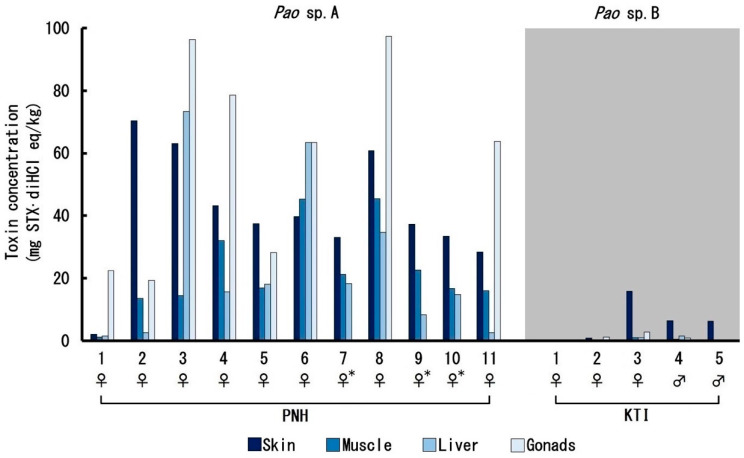
Toxin concentration in each tissue in each pufferfish collected from Phnom Penh (PNH, *Pao* sp. A) and Kratie (KTI, *Pao* sp. B). * The ovary was too small to analyze.

**Figure 6 toxins-12-00689-f006:**
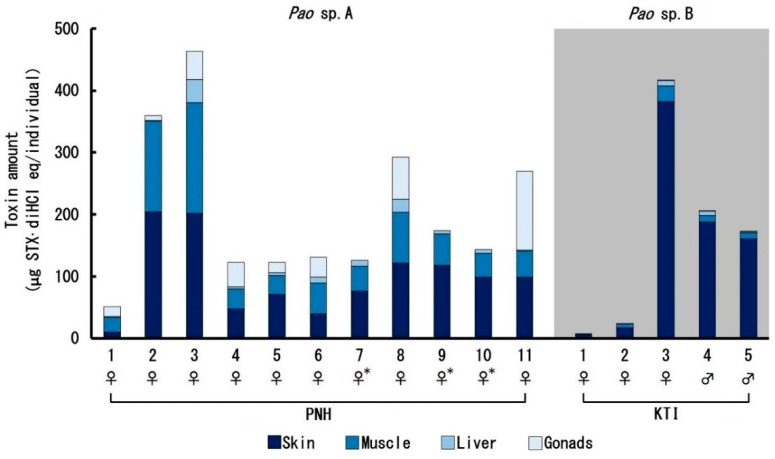
Total toxin amount per individual pufferfish collected from Phnom Penh (PNH, *Pao* sp. A) and Kratie (KTI, *Pao* sp. B). * The ovary was too small to analyze.

**Figure 7 toxins-12-00689-f007:**
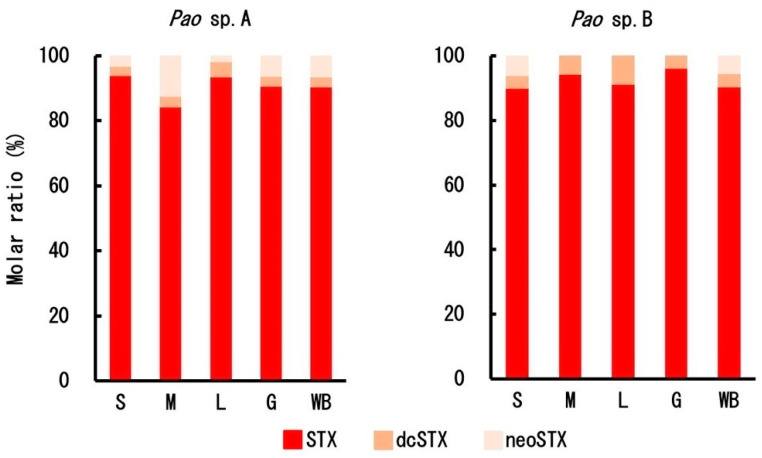
Toxin composition in each tissue in *Pao* sp. A (left) and *Pao* sp. B (right). S = skin, M = muscle, L = liver, G = gonads, and WB = whole body.

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
