# Peer review of "Phylogeny and Toxin Profile of Freshwater Pufferfish (Genus Pao) Collected from 2 Different Regions in Cambodia"

_toxins, 2020, doi:10.3390/toxins12110689_

Round 1

Reviewer 1 Report

The manuscript under reviewing provides phylogeny and toxin profile of Cambodian freshwater pufferfish. In general, the manuscript looks laconic and is suitable for publication in Toxins. Some minor concerns or suggestions You can see below.
Line 2: Set species/genus pufferfish name to the manuscript title
Line 24-25: Authors should change “Key Contribution” section. In Discussion section (Line 229-245). In Discussion section authors did not suggest that “TBT-bp2 isoforms” are involved in saxitoxin accumulation in freshwater pufferfish. Vice versa
Authors write, that a more comprehensive analysis must be conducted to provide a better understanding of the molecular mechanisms and evolutionary processes of TTX/STX accumulation in Tetraodontidae.
Line 38-40: I would like to pay attention on new publication Zang et al (doi:10.3390/toxins12050278). I think It will be good if Authors change this sentence according this new publication.

Author Response

Thank you very much for your useful comments. We revised our manuscript according to your comment (revised parts are indicated in red font).

Line 2: Set species/genus pufferfish name to the manuscript title
The genus name is now added to the title in the revised manuscript.

Line 24-25: Authors should change “Key Contribution” section. In Discussion section (Line 229-245). In Discussion section authors did not suggest that “TBT-bp2 isoforms” are involved in saxitoxin accumulation in freshwater pufferfish. Vice versa
Authors write, that a more comprehensive analysis must be conducted to provide a better understanding of the molecular mechanisms and evolutionary processes of TTX/STX accumulation in Tetraodontidae.
We do not agree with the reviewer’s comment. In Discussion section, we clearly claimed as “These findings together suggest that the TBT-bp2 gene could be one of the molecules involved in the kinetics (i.e., accumulation) of STX in the pufferfish body” (Line242-243 in the revised manuscript). The last sentence simply states that (although TBT-bp2 is suggested to be involved in STX accumulation) further research is needed to elucidate its mechanism and evolutionary processes.

 Line 38-40: I would like to pay attention on new publication Zang et al (doi:10.3390/toxins12050278). I think It will be good if Authors change this sentence according this new publication.
The publication of Zang et al. is now cited and the sentence “The accumulated TTX is gradually eliminated if supply via toxic food is not continued” is added (Line 41-42).

Reviewer 2 Report

The authors decribe the toxin content in two "species" of freshwater fish from Camabodia, genus Pao. Their attempt to identify the exact species using genetic barcoding provided mixed results, as no clear identity with related Pao species was achieved. Whether the specimens from the two locations are entirely new species or subspecies of already described species is not clear. 

The authors assayed the toxin content of the specimens and identified saxitoxin as the main component. However, in Fig 5 and 6 showing the individual toxin levels of the organs, the legends are not providing the essential information which column represents which organ. This needs to be included into the legends.

Line 57: it is stated that Pao specimens do not have a PSTBP gene (unpublished data). But why not presenting these data in the present paper? This wil explain why another potential toxin transporter, TBT-bp2 has been sequenced. Was the lack of the PSTBP gene in these specimens confirmed or does this statement apply to other Pao species? It is not convincing to state that this gene (and protein) is not generally present in Pao species, when it has´nt been investigated in the Cambodia specimens. This needs clarification.

Author Response

Many thanks for your valuable comments. We revised our manuscript according to your comment (revised parts are indicated in red font).

The authors describe the toxin content in two "species" of freshwater fish from Cambodia, genus Pao. Their attempt to identify the exact species using genetic barcoding provided mixed results, as no clear identity with related Pao species was achieved. Whether the specimens from the two locations are entirely new species or subspecies of already described species is not clear. 
The taxonomic status of the former genus Tetraodon (e.g. Pao) has been revised several times, and the Asian species were addressed into the genus Pao. Likewise, the status of the each Pao species seem to be still ambiguous, especially for those distributed in the Mekong basin across several countries, such as Pao leiurus. Moreover, the present manuscript also found even the molecular barcoding is suffering from inconsistency. Despite our maximum effort, the exact species was not identified for the specimens, but we were able to provide sufficient sequence data to identify whether a certain pufferfish specimen is Pao sp. A/Pao sp. B or not.

The authors assayed the toxin content of the specimens and identified saxitoxin as the main component. However, in Fig 5 and 6 showing the individual toxin levels of the organs, the legends are not providing the essential information which column represents which organ. This needs to be included into the legends.
The information which column represents which organ was shown under the graph (over the legends) in the previous manuscript. The indication is now modified to be larger and more prominent in the revised manuscript.

Line 57: it is stated that Pao specimens do not have a PSTBP gene (unpublished data). But why not presenting these data in the present paper? This will explain why another potential toxin transporter, TBT-bp2 has been sequenced. Was the lack of the PSTBP gene in these specimens confirmed or does this statement apply to other Pao species? It is not convincing to state that this gene (and protein) is not generally present in Pao species, when it hasn’t been investigated in the Cambodia specimens. This needs clarification.
The presence or absence of PSTBP gene was examined for Pao abei, Pao suvattii, and Leiodon cutcutia, which is a sister taxon to the genus Pao, based on genome and transcriptome (RNA-seq) analyses. We did not detect any partial sequence of the gene from both of the analyses. It is, thus, most likely that the whole genus Pao lacks the PSTBP gene. Meanwhile, the TBT-bp2 gene has been thought to encode a protein that binds xenobiotics such as TBT and has been identified as the substantial components of the PSTBP gene. Although an affinity of TBT-bp2 to STXs still remains unknown, these allow us to think that TBT-bp2 is a candidate molecule involved in TTX/STX accumulations. As the reviewer claimed, it might further explain the importance of TBT-bp2 in the STX accumulation if the lack of the PSTPB gene was shown here; however, it does not necessarily clarify the function of TBT-bp2 as a STX transporter as well. In addition, as mentioned above, our analyses were done for other specimens collected in Thailand but not in Cambodia, and we are currently preparing another manuscript on the analyses. This is the reason why we did not show the data.
At least following two points are now described in the revised manuscript (Line 58-61); 1) how we examined the absence of PSTBP, and 2) the examined samples are different from those of the present study.

Reviewer 3 Report

This manuscript describes the classification and the STX profiles of two kinds of freshwater pufferfish collected in Cambodia (designated PNH and KTI, respectively). The authors conducted phylogenetic analyses of partial sequences of mitochondrial genes, and demonstrated clearly that PNH and KTI are different species in the genus Pao. A partial sequence of the TBT-bp2 gene also differentiated the species as in the case of the mitochondrial genes. The toxin distribution profiles were investigated, showing the Pao sp. A and B contains STX and its analogs, especially high levels in the skin. The paper is well-written, and the authors have worked hard to produce a comprehensive dataset and detailed description of their research. The findings of this paper will be important for understanding the accurate toxin levels of freshwater pufferfish in Cambodia, and will contribute to creation of the food safety standard for the edible use of freshwater pufferfish. Some suggestion for revision of the text are listed below, I hope these comments will be helpful:

1. Page 4, line 104-105: Please consider conducting cloning experiments to determine the longer sequences of TBT-bp2 gene in KTI-3 and PNH-11, because the authors suggest TBT-bp2 gene could be one of the molecules involved in the kinetics of STXs in the Cambodian pufferfish. Thus, Acquiring longer sequences might certainly be of use for the next study. Otherwise omit KTI-3 and PNH-11 from Table A2 because I am not sure that those are necessary.

2. Page 5, Figure 4: It is enough to use Figure 3 to show the classification of these two species. Please consider moving Figure 4 to the Appendix.

3. Page 5, first column: Please insert the chromatogram of STX standards and that of samples (several representative samples are enough) into the Appendix for showing identification of the toxin.

4. Page 10, Table A1: Please revise the bold font of “Pao sp. A” and “PNH-1” as well as unnecessary underline.

Author Response

Many thanks for your useful comments. We revised our manuscript according to your comment (revised parts are indicated in red font).

1. Page 4, line 104-105: Please consider conducting cloning experiments to determine the longer sequences of TBT-bp2 gene in KTI-3 and PNH-11, because the authors suggest TBT-bp2 gene could be one of the molecules involved in the kinetics of STXs in the Cambodian pufferfish. Thus, acquiring longer sequences might certainly be of use for the next study. Otherwise omit KTI-3 and PNH-11 from Table A2 because I am not sure that those are necessary.
Although we considered cloning experiments to obtain longer or complete sequence of TBT-bp2 gene, PCR amplifications were not successful probably because of the presence of microsatellite-like sequences in an intron. This point is now added in the revised manuscript (Line 115-117). Table A2 is intended to clarify the difference of genotypes between Pao sp. A and Pao sp. B, and so we think it would be better to show all the data even for the specimens of which sequences were short or not obtained.

2. Page 5, Figure 4: It is enough to use Figure 3 to show the classification of these two species. Please consider moving Figure 4 to the Appendix.
Figure 3 shows the phylogenetic relationships among the Pao species including Pao sp. A and Pao sp. B based on a mitochondrial marker, while Figure 4 does not show their phylogenetic classification but the genetic relationships of TBT-bp2 gene, one of the nuclear gene, between Pao sp. A and Pao sp. B. Moreover, we could not necessarily expect the same results from mitochondrial gene analysis and nuclear gene analysis, and thus we do not think to move Figure 4 to the Appendix.

 3. Page 5, first column: Please insert the chromatogram of STX standards and that of samples (several representative samples are enough) into the Appendix for showing identification of the toxin.
Typical chromatograms have been inserted into the appendix.

 4. Page 10, Table A1: Please revise the bold font of “Pao sp. A” and “PNH-1” as well as unnecessary underline.
Table A1 has been revised as commented.